# pH-Sensitive Polyacrylic Acid-Gated Mesoporous Silica Nanocarrier Incorporated with Calcium Ions for Controlled Drug Release

**DOI:** 10.3390/ma15175926

**Published:** 2022-08-27

**Authors:** Jungwon Kong, Sung Soo Park, Chang-Sik Ha

**Affiliations:** 1Department of Polymer Science and Engineering, School of Chemical Engineering, Pusan National University, Busan 46241, Korea; 2Division of Advanced Materials Engineering, Dong-Eui University, Busan 47340, Korea

**Keywords:** mesoporous silica, polyacrylic acid, calcium ion, 5-fluorouracil, drug delivery

## Abstract

In this work, polyacrylic acid-functionalized MCM-41 was synthesized, which was made to interact with calcium ions, in order to realize enhanced pH-responsive nanocarriers for sustained drug release. First, mesoporous silica nanoparticles (MSNs) were prepared by the sol-gel method. Afterward, a (3-trimethoxysilyl)propyl methacrylate (TMSPM) modified surface was prepared by using the post-grafting method, and then the polymerization of the acrylic acid was performed. After adding a calcium chloride solution, polyacrylic acid-functionalized MSNs with calcium-carboxyl ionic bonds in the polymeric layer, which can prevent the cargo from leaking out of the mesopore, were prepared. The structure and morphology of the modified nanoparticles (PAA-MSNs) were characterized by X-ray diffraction (XRD), Fourier-transform infrared (FT-IR) spectroscopy, transmission electron microscopy (TEM), and N_2_ adsorption–desorption analysis, etc. The controlled release of guest molecules was studied by using 5-fluorouracil (5-FU). The drug molecule-incorporated nanoparticles showed different releasing rates under different pH conditions. It is considered that our current materials have the potential as pH-responsive nanocarriers in the field of medical treatment.

## 1. Introduction

Responsive materials have attracted a lot of interest in research and industry in recent years. It can provide proper opportunities in many fields such as the automotive [1], aerospace [2,3], electric device [4], and medical application [5,6,7] industries. In particular, this smart material’s sensitivity to specific stimuli can show positive possibilities for advances in cancer therapy [8,9,10,11]. Cancer is a disease caused by abnormal cell growth and is still the main cause of death in humans. There are several treatments available to treat cancer, but chemotherapy is one of the most commonly used cancer treatments [12]. However, present chemotherapeutic drugs have low specificity to targeted cancerous cells [13,14]. When a drug is administered to the human body, the lack of specificity of a drug molecule leads to a high-dosage regimen and causes undesired interactions of a drug with normal cells [15]. This can have a detrimental effect on a person, causing damage to healthy organs or tissues. The efficiency of treatment would be decreased and side effects will follow related to systemic toxicity. To overcome this disadvantage, many researchers have been investigating the controlled drug-delivery system (CDDS) [16,17]. Stimuli-responsive materials such as polymers [18,19], hydrogels, and many kinds of inorganic nanoparticles [20,21] can be actively utilized in CDDS as a means of transport. The released amounts of drugs can be controlled in response to external stimuli (temperature, light irradiation, pH, redox, etc.). In the case of cancer treatment, pH-responsive materials are gaining lots of interest because cancer tumor tissue is more acidic than normal tissue. (pH 4.0~6.0) [22]. The pH-sensitive drug delivery system, which can allow for the selective release at a low pH region and no response to the normal physiological environment, would be a worthy method of treatment [23,24,25].

Among various drug delivery systems, mesoporous silica nanoparticles (MSNs) have been the subject of interest in recent years because of their large surface area, high pore volume, tunable pore size, and chemical stability [26,27]. It also has great biocompatibility and high loading capacity among other systems [28]. By adding a drug or other cargo inside the MSNs, it can protect them from degradation and guard them against leakage. Also, MSNs can be functionalized easily on the internal and external surface so that diverse modifications are possible to prevent premature abruption of drugs or cargos from the pores of mesoporous silica particles [29]. 

The pore walls or the surfaces of MSNs were functionalized with organic moieties such as amine, carboxyl, thiol, and their derivatives to be used as drug carriers [30,31,32,33,34,35,36]. Moreover, after loading various drugs with hydrophilic or hydrophobic properties, DNA, enzyme, etc., the properties of drug-delivery materials, including controlled release properties and toxicity to cancer cells, were studied by many researchers [30,31,32,33,34,35,36]. Polymers and small molecules were used as the functional groups for the controlled release of drug molecules. The functionalized and drug-loaded MSNs exhibited the controlled release behavior of drug molecules by various stimuli such as pH, temperature, light, enzyme, redox, etc. [30,31,32,33,34,35,36]. In the case of most mesoporous silica loaded with the anticancer drug 5-FU, the controlled release of 5-FU was induced by pH stimuli [37,38,39,40,41].

From the perspective of the drug-delivery material, polymers with adjustable properties such as biodegradability and biocompatibility have been studied a great deal as materials for drug-delivery systems [28,42,43]. Phase-transfer polymers are sensitive to external stimuli and are continuously or discontinuously hydrated [44,45]. These polymers show a change in physical properties of external stimuli that include chemical or biochemical stimuli like pH and metabolites, and physical stimuli such as temperature, light electric field, and solvency [46,47]. However, drug-polymer conjugates have bad stability, and loading capacities are low [28]. To optimize this transport platform, we have to address these issues. 

By functionalizing polymers on the surface of the mesoporous silica nanoparticles, we can obtain a hybrid nanoparticle that has high stability and loading capacity as well as adjustable properties, which can make sensitive stimuli-switchable nanocarriers [26,48,49,50,51]. The connection of polymers such as polyacrylamide [52], poly(methacrylic acid) [53,54], poly(diethylaminoethyl methacrylate) [55], chitosan [56,57,58], poly(4-vinylpyridine) [50,59] and others [60] realized the pH-responsive polymer-coated mesoporous silica particles for drug-delivery systems [61].

In general, it is known that the size of nanoparticles of about 100 nm is most suitable for delivery to cells by endocytosis [62]. However, it has been reported that mesoporous silica with a large particle size can also be sufficiently applied as a carrier for drug delivery in cells. Moorthy et al. [37] reported on the research of the simultaneous cell imaging and the anticancer drug delivery in MCF-7 cells with the safranin–diurea bridged and 5-fluorouracil (5-FU)-loaded mesoporous organosilica (5-FU loaded SDU-HMS). The 5-FU-loaded SDU-HMS showed excellent cytotoxicity to the MCF-7 cells with cell viability of approximately 18%. The size of mesoporous silica was in the range of 200–800 nm. Moorthy et al. [63] also reported the synthesis of the carboxyl- (DU-MSH-COOH) and amine-functionalized mesoporous silica (DU-MSH-NH_2_) with a particle size in the range of 500 nm–1.5 μm. The internalized 5-FU loaded DU-MSH-COOH and DU-MSH-NH_2_ into MCF-7 cells and showed excellent cytotoxicity of 76% and 82%, respectively. Tao et al. [64] reported on the endocytosis and the time-dependent enhanced cytotoxicity of anticancer platinum drug-loaded MCM-41 (particle size in the range of 500–900 nm) or SBA-15 (particle size of the micrometer). The mesoporous silica particles with large sizes may be suitable for the cells with comparatively large sizes such as MCF-7 and HeLa cells [37].

Meanwhile, calcium ions play important roles inside and outside cells in the human body. Extracellular calcium is not only a major component of cartilage and skeleton, but also plays a role in excitatory contraction in the heart and muscle, synaptic transmission in the nervous system, platelet activation and coagulation, and extracellular excretion, and is also involved in the secretion of various hormones. Intracellular calcium plays an important role as a signal transmitter in cell division, muscle contraction, cell migration, membrane transduction, secretion, etc. [65,66]. 

To the best of our knowledge, drug-delivery systems of 5-FU based on mesoporous silica and modified with functional groups that can be used as gatekeepers by controlling the interaction between polymer (PAA) and metal ions (Ca^2+^) by acidity have rarely been reported. In this work, 5-FU encapsulated mesoporous silica was prepared, which has polyacrylic acid (PAA)-Ca^2+^ complexes on the surface as a gatekeeper. An anti-cancer drug, 5-FU can be used in the treatment of stomach cancer, pancreatic cancer, and so on [67]. It is negatively charged and has low lipid solubility. First, MCM-41 nanoparticles were synthesized by a liquid crystal templating mechanism. Secondly, PAA was attached to the surface of MCM-41. PAA has low toxicity and provides more hydrophilicity to mesoporous silica materials [68]. Furthermore, it has a carboxylic group that can make electrostatic interaction with calcium ions. By mixing calcium chloride solution with the 5-FU loaded MCM-41-PAA, metal ion complexes were formed by complexation between the calcium ions and carboxyl groups. That can be a barrier in a neutral and basic environment by capping the pore exits. However, in an acidic environment, the Interactions between carboxylate ions and calcium ions would be decreased and the release of drugs inside the pore could be possible. Preparation of a drug delivery nanomatrix is presented, and characterization of each step was carried out. As-made MSNs have a large surface area and pore volume that allow them to be used properly as drug carriers. The release characteristics of 5-FU were also assessed. It showed selective release amount under each different pH environment. The results indicated that its pH responsivity is sufficiently available to apply in cancer treatment.

## 2. Experimental Section

### 2.1. Materials

Cetyltrimethylammonium bromide (CTABr, >99%), tetraethyl orthosilicate (TEOS, 98%), 3-(trimethoxysilyl)propyl methacrylate (TMSPM), toluene (anhydrous, ≥99.8%), acrylic acid, ethanol (anhydrous), potassium persulfate (KPS, ≥99%), N,N′-methylenebis(acrylamide) (MBA), calcium chloride and 5-fluorouracil (5-FU, ≥99%) were used. All the above materials were purchased from Sigma-Aldrich. Sodium hydride (bead, 98.0%) (NaOH) and dichloromethane was purchased from Samchun Chemical. Hydrochloric acid (35%) (HCl) was purchased from Matsunoen Chemicals Ltd. All the reagents were used without further purification.

### 2.2. Synthesis of MCM-41

MCM-41 nanoparticles were synthesized according to the method reported in the literature [69]. CTABr (1 g, 2.74 mmol) and NaOH (0.28 g, 7 mmol) were added to deionized (DI) water (480 g, 26.6 mol) in a 1 L round bottom (RB) flask and stirred at room temperature for 1 h to obtain a homogeneous mixture. The homogeneous mixture was heated for 30 min to achieve a stabilized temperature (80 °C) with constant stirring. Then TEOS (5 mL, 22.39 mmol) was added dropwise to the solution and the total mixture was stirred continually at 80 °C for two more hours. The molar composition of the total mixture was 1.0 SiO_2_:0.12 CTABr:0.31 NaOH:1188 H_2_O. The obtained product was isolated, washed with ethanol several times by centrifugation, and dried in an oven at 70 °C for 24 h. The synthesis procedure was briefly illustrated in Figure 1. The resulting product was named as ‘As-synthesized MCM-41’.

### 2.3. Functionalization with Double Bond Groups (MCM-41-TMSPM)

For the removal of water bonded with silanol groups on the surface of nanoparticles, as-synthesized MCM-41 was degassed at 110 °C for more than 12 h under vacuum. A total of 2 g of the degassed as-synthesized MCM-41 was transferred to a 100 mL round bottom flask with 50 mL of anhydrous toluene. After stirring for 1 min at room temperature, 2 mL of TMSPM was added to the system. The mixture was kept at 110 °C for 24 h with constant stirring under a nitrogen atmosphere. Then the mixture was centrifugated 6 times with toluene and DCM. The resulting product was named as “as-MCM-41-TMSPM”. After this procedure, solvent extraction was conducted to remove CTABr surfactant in the mesopores. A total of 1.5 g of as-MCM-41-TMSPM was stirred for 12 h in 225 mL of ethanol solution containing 35 wt.% HCl (4.5 mL) at 60 °C. This procedure was repeated 3 times. The obtained product was isolated and washed with toluene several times by centrifugation and dried in an oven at 70 °C for 24 h. The synthesis procedure was briefly illustrated in Figure 1. The resulting product was named as “MCM-41-TMSPM”.

### 2.4. Synthesis of Polyacrylic Acid-Functionalized MCM-41 (MCM-41-PAA)

The functionalization of polyacrylic acid was performed by referring to the process of literature [70]. Polyacrylic acid-functionalized MCM-41 was prepared by free-radical polymerization of acrylic acid (AA). A total of 0.46 g of MCM-41-TMSPM was degassed at 70 °C for 12 h under vacuum conditions. The degassed sample was dispersed in 40 mL of deionized (DI) water: ethanol mixture (1:1 *v*/*v*) in a 250 mL round bottom flask and purged with nitrogen for 6 h for the removal of oxygen in the mesopores of MCM-41-TMSPM. In this mixture, 1.38 g of AA in 40 mL DI water: ethanol mixture (1:1 *v*/*v*) was added with 12.3 mg of N,N′-methylenebis(acrylamide) (MBA) as a crosslinker agent. The mixture was stirred for 1 h at 80 °C under a nitrogen atmosphere. After that, 24 mg of potassium persulfate (KPS) was added as an initiator agent and the total mixture was stirred for 12 h at 80 °C under a nitrogen atmosphere. The obtained product was filtered by centrifugation and washed with water, ethanol, and dichloromethane to remove the polymer physically attached to the surface of nanoparticles. The synthesis process was briefly summarized in Figure 1. The resulting product was named as ‘MCM-41-PAA’.

### 2.5. Preparation of Calcium Ion Interacted 5-FU Loaded MCM-41-PAA Matrix (Ca@MCM-41-PAA_5-FU)

The chemotherapy drug 5-FU was loaded into the MCM-41-PAA by adding the MCM-41-PAA (100 mg) in an aqueous solution of 5-FU (10 mL, 1 mg mL^−1^) at room temperature and stirred for 24 h. Then CaCl_2_ solution (10 mL, 0.01 M) was added and the reaction mixture was stirred for another 24 h, leading to the formation of electrostatic interactions between Ca^2+^ and the COO^−^ at a pH 7.4 environment. The obtained mixture was isolated by centrifugation and washed with DI water two times. The resulting materials were dried in an oven at 70 °C for 12 h. The resulting product was named as “Ca@MCM-41-PAA_5-FU”. The loading of 5-FU in MCM-41-PAA nanocarrier was confirmed by wide-angle X-ray diffraction (WXRD) and energy-dispersive X-ray spectroscopy (EDS) and the loading amount of 5-FU was evaluated by UV-vis spectroscopy. The synthesis procedure was briefly displayed in Figure 1.

The 5-FU loaded MCM-41 (named as ‘MCM-41_5-FU’) and 5-FU loaded MCM-41-PAA (named as “MCM-41-PAA_5-FU”) without Ca^2+^ ions were prepared by the same method that was previously mentioned.

### 2.6. Drug Releasing Test

A total of 20 mg of MCM-41_5-FU, MCM-41-PAA_5-FU, and Ca@MCM-41-PAA_5-FU was added to 5 mL of PBS (phosphate buffer saline) solution and the dispersion was filled into a dialysis bag (cutoff molecular weight (Mw) = 1.2 kDa). The bag with dispersion was immersed in 20 mL of PBS solution of different pH (5.4, 7.4) at 37 °C in a shaking incubator. At the specific time intervals, 2 mL of external medium was taken and replaced with the same volume of a new PBS buffer solution immediately. The amount of released 5-FU was calculated by measuring intensity by UV-vis spectrophotometer at 266 nm.
$$\%R_{t}=\frac{C_{t}\cdot V_{1}+V_{2}\cdot \left(C_{t-1}+C_{t-2}+\cdots +C_{0}\right)}{W_{0\;}\cdot L}\times 100\%$$

The above equation was applied to calculate the drug release degree of samples. In this equation, *C_t_* is the drug concentration at time interval *t*, *C*_*t*−1_ + *C*_*t*−2_ are drug concentrations prior to time interval *t* (*C*_0_ = 0), *V*_1_ is the total volume of the release bath (25 mL), and *V*_2_ is the volume extracted for UV-vis analysis (2 mL). *W*_1_ is the initial weight of the 5-FU-loaded MSNs, and L is the drug-loading capacity of the MSNs_5-FU [38]. Also, calibration curves of 5-FU in DI water, and PBS buffer at different pH (pH 5.4, 7.4) were used to translate UV-vis intensity to the concentration of 5-FU in the medium.

### 2.7. Measurements and Characterization

X-ray powder diffraction (XRD, Bruker AXS) was performed by using Cu-Kα radiation (1.5418 Å) at 30 kV and 15 mA in the 2θ range, 0.5~10° (LXRD), 10~80° (WXRD). The characteristic surface functional groups were analyzed by Fourier transform infrared spectroscopy (FTIR, JASCO (FTIR-4100), Hachioji, Japan) at a scanning range of 400–4000 cm^−1^ in KBr. Transmission electron microscopy (TEM, TALOS F200X) was performed at an accelerating voltage of 200 kV and energy dispersive X-ray spectroscopy (EDS) fast signal detection was conducted. High-resolution low-voltage scanning electron microscopy (HRLV-SEM, JSM-7900F) was conducted at an operating voltage of 2.0 kV. Thermogravimetric analysis (TGA, TA, Q50) was carried out at a heating rate of 10 °C min^−1^ in nitrogen gas from 30 °C to 800 °C to measure the thermal stability of the samples. The adsorption/desorption isotherms of nitrogen at 77 K were measured by using a Nova 4000 e surface area and pore size analyzer. Before starting the measurements, each sample was outgassed for 12 h under vacuum conditions at 343 K. The pore surface area was calculated by Brunauer Emmet-Teller (BET) method and the pore size distribution curve was obtained from an analysis of the adsorption branch by using the Barrette–Joynere–Halenda (BJH) method. The zeta potentials were observed by Zetasizer Nano ZS (Malvern Instruments Ltd., Malvern, UK) at room temperature. The particle size distribution curves were also measured by using a dynamic light-scattering spectrophotometer (Zetasizer Nano, Malvern, UK). Ultraviolet-visible (UV-vis) spectrophotometry (Hitachi U-2010) was used to measure the drug release rate.

## 3. Results and Discussion

### 3.1. Characterization of MCM-41 and Functionalized MCM-41

#### 3.1.1. Scanning Electron Microscopy (SEM) and Transmission Electron Microscopy (TEM)

Figure 1 shows SEM images of (a,e) MCM-41, (b,f) MCM-41-TMSPM, (c,g) MCM-41-PAA, and (d,h) Ca@MCM-41-PAA_5-FU. The images (Figure 1a,e) of the pristine MCM-41 exhibited a homogenous spherical morphology of particles with an average size of around 140 nm. After the functionalization of polyacrylic acid (PAA) on the surface (Figure 1c,g), the nanoparticles maintained their morphological integrity. The surface of the nanoparticles looks uneven which is presumed to be due to the grafting of polyacrylic acids. After the incorporation of calcium ions into the MCM-41-PAA (Figure 1d,h), the complexation of carboxyl functional groups and calcium ions also made the surface of the particle roughen.

Figure 2 shows TEM images of samples (a–d). All the modified and unmodified MCM-41 have well-organized, periodic, hexagonal mesopore arrays and present channel structures with parallel stripes. This proves that the ordered 2D hexagonal mesostructure was obtained in this work [71]. After the polymerization of PAA (Figure 2c), some grafting of the polymer can be seen on the surface of MCM-41. In the TEM images of Figure 2d, one can observe the coating of polymeric layers on the surface but no significant difference from the prior step’s image (Figure 2c). The existence of calcium ions by the incorporation with carboxylate functional groups can be measured by the method of EDS mapping, which will be mentioned later. The structure without loss of well-arranged mesopore arrays in a long range after modification steps is preferred for loading guest molecules (Figure 2c,d) [72].

#### 3.1.2. Elemental Analysis

Figure 3 shows FE-TEM image (a), EDS mapping (b–h), and spectrum (i) of Ca@MCM-41-PAA_5-FU. The element Ca was observed to be uniformly dispersed on the MCM-41-PAA nanoparticles (Figure 3h). Furthermore, the encapsulation of 5-FU can be proven by the presence of elements N and F in the nanostructure which comes from the component of 5-FU molecule (Figure 3c,d). Other elements such as Si, O, and C also can be confirmed. In particular, distinct dispersion of C elements around the sphere nanoparticle indicates PAA polymer coating was synthesized successfully. In addition, EDS spectra of Ca@MCM-41_5-FU demonstrate the weight percentage of Ca (0.73 wt.%), N (1.94 wt.%), and F (0.53 wt.%) in the polymer functionalized MCM-41 nanocarrier.

#### 3.1.3. X-ray Diffraction (XRD)

Low-angle X-ray diffraction (LXRD) patterns of MCM-41, MCM-41-TMSPM, MCM-41-PAA, and Ca@MCM-41-PAA_5-FU are shown in Figure 4. MCM-41 presents intense reflections of d-spacing (100) and three additional peaks with low intensities at (110), (200), and (210), which correspond to the values of 2θ = 2.1°, 3.8°, 4.4°, and 5.7°, respectively (Figure 4a). These peaks can indicate the organized structure of samples associated with a p6mm hexagonal symmetry of the ordered MCM-41-type materials [73]. As the modification of TMSPM (Figure 4b) and functionalization of PAA (Figure 4c) proceeded, the intensities of d-spacing (110), (200), and (210) were successively decreased, which is proof of the success of the synthesis procedure without a loss of mesoporous structural ordering. Moreover, broadening and slight shifts to the higher angle of d-spacing (100) and (200) occurred due to the functionalization of PAA on the mesoporous silica [74]. The incorporation of 5-FU into the Ca@MCM-41-PAA_5-FU can be established by the significantly decreased intensity of LXRD peaks at d-spacing (100), (110), and (200) (Figure 4d). The wide-angle X-ray diffraction (WXRD) pattern of Ca@MCM-41-PAA_5-FU (Figure 5d) compensates for this fact. The WXRD patterns were measured to check up whether the crystalline phase of 5-FU molecules was shown on the surface of the pristine MCM-41 and the functionalized mesoporous silica nanoparticles. On the WXRD pattern of 5-FU, a characteristic strong peak at 2θ = 28° is observed (Figure 5a) [75]. In the case of 5-FU-loaded nanoparticles such as MCM-41_5-FU, MCM-41-PAA_5-FU, and Ca@MCM-41-PAA_5-FU, broad diffractions peak at 2θ values between 17° and 36° with very low intensities, which are typical for the amorphous state of silica structure [74] (Figure 5b–d). These patterns without specific peaks of 5-FU imply that the total incorporation of 5-FU into the mesopores was well proceeded with no existence of 5-FU cluster on the outer surface.

#### 3.1.4. Thermogravimetric (TG) Analysis

By the measurement of TGA analysis (Figure 6), one can observe the extent of functionalization of each proceeding step and loading of guest molecules (5-FU) can be observed in mesoporous nanoparticle samples. The weight percentages after heating up to 800 °C with a rate of 10 °C min^−1^ in a nitrogen atmosphere of MCM-41, MCM-41-TMSPM, MCM-41-PAA, and Ca@MCM-41-PAA_5-FU were 89.9%, 80.1%, 70.1%, and 65.2%, respectively. The initial weight loss under 200 °C is caused by thermo-desorption of physisorbed water or residual solvents [70]. Ca@MCM-41-PAA_5-FU probably contains a higher amount of H_2_O than other samples because it contains calcium ions and carboxyl groups with high hydrophilicity. Meanwhile, 5-FU may exhibit weight loss due to some decompositions as it is heated up to 200 °C. These factors will lead to a significant weight loss of Ca@MCM-41-PAA_5-FU below 200 °C as compared with other samples. A decrease of weight from 200 to 800 °C was assigned to the loss of organic functional groups on the mesopore walls. During this process, water molecules may also form through the silanol condensation in the pore walls [76]. In the case of MCM-41 (Figure 6a), the degradation of the remaining CTAB surfactant (from 180 to 340 °C) and the condensation of silanol groups (Si-OH) (up to 800 °C) in the frameworks can be occurred resulting in 10.1% weight loss [77]. Likewise, the total weight loss of the samples was 19.9% for MCM-41-TMSPM (Figure 6b) and 29.9% for the MCM-41-PAA (Figure 6c) due to the decomposition of TMSPM and PAA organic functional groups, respectively. Moreover, 34.8% weight loss of Ca@MCM-41-PAA_5-FU can be proof of loading of 5-FU drugs inside mesopores (Figure 6d). The increase of weight loss implies that a much higher extent of functionalization with the organic groups or loading of 5-FU was proceeded at this time in steps [78].

#### 3.1.5. Fourier Transform Infrared (FTIR) Spectroscopy

Figure 7 shows FTIR spectra which make it possible to evaluate the surface functionalization of mesoporous silica nanoparticles. In the FTIR spectrum of MCM-41 as shown in Figure 7a, the intense peak at 1082 cm^−1^ appeared as a Si–O–Si antisymmetric stretching vibration. Besides this peak, the peaks at 959, 797, and 459 cm^−1^ ascribed to the bending vibration of the Si–OH, symmetric stretching vibration of the Si–O–Si and the stretching vibration of Si–O, respectively [79]. All these peaks are characteristic adsorption peaks of mesoporous silica (SiO_2_). The broad peaks at 3419 cm^−1^ and absorption peaks at 1629 cm^−1^ in this sample appeared due to the OH stretching and H–O–H bending vibration adsorbed water. In the case of MCM-41-TMSPM (Figure 7b), additional peaks at 2953 and 2892 cm^−1^ correspond to the asymmetric and symmetric C–H stretching, which can be proof of successful modification of TMSPM on the surface. Likewise, adsorption peaks of ester C=O stretching and C=C stretching vibration of TMSPM molecule structure are measured at 1704 and 1635 cm^−1^, respectively [80]. After functionalization of PAA (Figure 7c), one can observe the carbonyl C=O stretching vibration peak of carboxylic acid at 1715 cm^−1^, which has a higher intensity than the C=O stretching adsorption peak of TMSPM. In addition, the peak at 1402 cm^−1^ was obtained due to the symmetric O–C–O stretching vibration of the carboxyl groups [81]. In the spectrum of Ca@MCM-41-PAA_5-FU (Figure 7d), O–C–O stretching vibration peaks of carboxylate groups were observed at 1568 and 1412 cm^−1^. The peaks were slightly shifted due to the interaction between calcium ions and carboxyl groups compared to the MCM-41-PAA without calcium ions [82].

#### 3.1.6. N_2_ Adsorption–Desorption Isotherms and Pore Size Distributions

N_2_ adsorption–desorption isotherms (Figure 8A) and pore-size distributions (Figure 8B) of MCM-41, MCM-41-PAA, and Ca@MCM-41-PAA_5-FU are shown in Figure 8. Table 1 lists the surface area, pore volume, and pore size of MCM-41, MCM-41-PAA, and Ca@MCM-41-PAA_5-FU. Among various kinds of isotherms, Type IV was gained because of the structure of the mesopore [83]. Type IV isotherms appear by capillary condensation, and they have a hysteresis loop because of the difference between their monolayer–multilayer adsorption and desorption characteristics [83]. In the case of MCM-41, it has a high surface area (900 m^2^ g^−1^), large pore volume (0.85 cm^3^ g^−1^), and pore size (2.7 nm), which is sufficient to be used as the functionalized nanocarrier of guest molecules. After polymerization of acrylic acid, the surface area, pore volume, and pore size were decreased due to the presence of functional groups of PAA that partly block the adsorption of nitrogen molecules on the surface. The decrease of these parameter values indicates the polymerization of acrylic acid proceeded well and the PAA groups covered the wall of nanoparticles. Moreover, the decline of surface area, pore volume, and pore size of Ca@MCM-41-PAA_5-FU compared to MCM-41-PAA shows that the loading of 5-FU drugs and calcium ions occurred in the mesopores. In addition, it appears Type I isotherm characteristics on Ca@MCM-41-PAA_5-FU which can be attributed to partial loading of 5-FU drugs and some of the calcium ions in mesopores that hinder steep capillary condensation [84]. Meanwhile, the pore size of Ca@MCM-41-PAA_5-FU was too small to measure (<2.0 nm).

#### 3.1.7. Zeta Potential

As shown in Table 2, zeta potential measurements were carried out to figure out the surface charge of MCM-41, MCM-41-TMSPM, MCM-41-PAA, MCM-41_5-FU, MCM-41-PAA_5-FU, and Ca@MCM-41-PAA_5-FU in aqueous suspensions. In the case of MCM-41, the massive number of silanol groups on the surface proceed to the negative zeta potential value of −18.3 mV assuring that the negative charges (Si-O^−^) were generated from deprotonated silanol groups [85]. After modification of TMSPM, the zeta potential value of MCM-41-TMSPM is shifted to −0.135 mV due to the terminal alkenyl groups which have relatively more neutral charges than that of silanol groups [86]. Subsequent functionalization of PAA led zeta potential values to decrease up to −51 mV, as carboxyl groups (−COOH) are deprotonated as negatively charged carboxylate groups (−COO^−^). This distinct change of zeta potential value verifies the existence of a significant amount of carboxylic acid groups on the nanoparticle surface which are positioned by the functionalization of PAA [87]. Likewise, the zeta potential was measured after loading 5-FU into the MCM-41, MCM-41-PAA, and calcium-incorporated MCM-41 samples. In the case of MCM-41_5-FU, the early burst of 5-FU in an aqueous solution may affect the surface charges of MCM-41 by forming hydrogen bonds between silanol groups and 5-FU which has N and F elements in molecule structure [88]. For this reason, the zeta potential value changes to 0.132 mV. Otherwise, the value of 5-FU-loaded MCM-41-PAA (MCM-41-PAA_5-FU) does not have a significant difference from that of MCM-41-PAA. It might be less affected by the release of drugs because the release amount from MCM-41-PAA_5-FU is less than that from MCM-41_5-FU. In addition, the massive amount of carboxyl acid groups in PAA chains can compensate for the variable factor-changing zeta potential of the nanoparticle surface. The sample loaded with both 5-FU and calcium ion (Ca@MCM-41-PAA_5-FU) has a zeta potential value of 0.0108 mV. This zeta potential value shows that the interaction between carboxyl groups and calcium ions was formed, resulting in a negatively charged surface of MCM-41-PAA that would be neutralized in aqueous suspension.

#### 3.1.8. Dynamic Light Scattering (DLS)

The hydrodynamic diameters of MCM-41 (Figure 9A(a)) and MCM-41-PAA (Figure 9A(b)) can be obtained in aqueous solutions by DLS analysis (Figure 9A). The particle diameter of MCM-41 and MCM-41-PAA was 267.4 and 335.0 nm, respectively. These sizes are much bigger than those measured by SEM because of the existence of nonuniformly dispersed particles and the formation of a hydrated layer in water [89]. The increase in particle size of MCM-41-PAA compared to MCM-41 was made by the existence of hydrophilic PAA chains on the surface of MCM-41 and the robustness of adhesion between MCM-41-PAA nanoparticles than MCM-41 [90]. Moreover, the particle size of Ca@MCM-41-PAA was observed at different pH environments in PBS buffer to compare the pH-responsive behavior of the polymer-metal ion complex layer (Figure 9B). The hydrodynamic diameter was 394.0 nm at pH 5.4 (Figure 9B(a)) and 349.7 nm at pH 7.4 (Figure 9B(b)), respectively. The decrease of diameter at higher pH (7.4) implies that the contraction of polymer chains occurred by the interaction with calcium ions.

#### 3.1.9. Drug Releasing Test

Figure 10 shows the in vitro 5-FU-releasing profiles of 5-FU-loaded MCM-41 ((A) MCM-41_5-FU), 5-FU-loaded MCM-41PAA ((B) MCM-41-PAA_5-FU), and 5-FU-loaded MCM-41-PAA encapsulated by polymer-calcium ion complex ((C) Ca@MCM-41-PAA_5-FU) at (a) pH 5.4 and (b) 7.4 for 72 h, respectively. As shown in Figure 10A, the burst release of 5-FU was observed, and the total release amount was as high as close to 90~95%. This result is attributed to the dissociation of physically adsorbed 5-FU with silanol groups on the surface of MCM-41. Meanwhile, it shows a higher release amount at a pH 5.4 environment than that of pH 7.4 due to the partial protonation of 5-FU (pKa = 8.2) and silanol groups on the surface of MCM-41 that causes electrostatic repulsion of drug molecules under lower pH condition (pH 5.4) [63]. At a neutral pH condition (pH 7.4), both hydrophilic 5-FU and silanol groups made a hydrophilic–hydrophilic interactions and hydrogen bonding could be formed so that the release rate of 5-FU drugs could be slower [63]. MCM-41_5-FU showed a release amount of approximately 67% within a short time of 1 h (Figure 10A). The result can be due to the fact that drug molecules present outside and at the entrance to the mesopores are rapidly released. After that time, the drug is released slightly slowly until the drug release time of 24 h, and this result may be attributed to the route of release from the inside of the mesopores to the outside.

In the case of 5-FU-loaded MCM-41-PAA (MCM-41-PAA_5-FU) (Figure 10B), it shows a significantly high total release amount in pH 7.4 (96.6%) compared to pH 5.4 (42.5%). In a neutral pH environment (pH 7.4) (Figure 10B(b)), carboxylic acid groups can alter their ionization/deprotonation behavior depending on the pH environment. Most of the carbonyl groups of polyacrylic acid (pKa = 4.8) are negatively charged at pH 7.4 which makes electrostatic repulsion between polymer chains. In addition, its high solubility results in PAA chains’ swelling behavior. The open pore state caused by this behavior of the polymer allows 5-FU to migrate to the exit of the pore and release into the solution [91]. Note that there is no significant interaction between 5-FU (pKa = 8.2) and carboxyl groups [63]. In contrast, in a pH 5.4 environment (Figure 10B(a)), the contraction of PAA chains is contributed by hydrogen bonding between carboxyl groups by protonation. The result could block the pore entrances so that the release of 5-FU would be hindered [91]. Meanwhile, it showed a rapid release behavior of 5-FU with a short time of less than 1 h. This result may be attributed to the release of drug molecules present outside and at the entrance to the mesopore. A similar release behavior was also shown in Figure 10C.

Compared to the results of Figure 10B, in Figure 10C, the tendency of the drug release amount was changed. The total release amount in pH 5.4 and pH 7.4 was 93.7% and 54.1%, respectively. In a neutral environment (pH 7.4) (Figure 10C(b)), the carboxylate groups with negative charges could form COO^−^–Ca^2+^ electrostatic interaction. This ionic bond makes a polymer-metal ion complex layer around MCM-41 nanoparticles, so the release of 5-FU would be obstructed. At pH 7.4, however, some of the 5-FU molecules can be deprotonated [92], and then the interaction between 5-FU and calcium ions can be formed that can break the COO–Ca^2+^ interaction. This disassociation of polymer-metal ion bonds could arise from the leakage of 5-FU drugs contained in mesopores. At pH 5.4, the interacted polymer-calcium ion layer was disbanded and the drug release amount increased. The protonation of carboxyl groups at comparably low pH leads to the withdrawal of the electrostatic interaction between COO^−^ and Ca^2+^. This disruption of the polymer-metal ion layer led to the obvious release of 5-FU drugs into the solution. Also, Cl^−^ ions in PBS buffer support the formation of calcium chloride and replace carboxylate groups, thereby disrupting the ionic interaction between COO^−^ and Ca^2+^ [93]. By these results, the Ca^2+^ interacting polymeric layer can act as a pH-selective drug-release barrier and inhibit burst release in a normal physiological environment.

In a mildly acidic environment (pH 5.4), the polymeric gate of Ca@MCM-41-PAA_5-FU helped to allow the controlled release of 5-FU and Ca^2+^ ions. Meanwhile, at pH 7.4, the gate of nanoparticles remained mostly closed, and the drugs were stored because of the maintenance of the calcium ion-polymeric layer. If this nanocomplex got into the cancer cell by endocytosis, the interaction between the calcium ion and polymer would be destroyed in a low pH cytoplasm system with a higher concentration of Cl^−^, which can raise the higher release of 5-FU drugs [66]. Consequently, we can predict that this gating mechanism has a possibility to be applied to various medical treatments by its specific pH responsiveness for 5-FU release. Furthermore, this stimuli-responsive property of the synthesized material shows potential as an effective delivery system for many applications in the industry.

The Korsmeyer–Peppas model is used to describe and analyze the release of a drug from a polymeric nanoparticle dosage form such as a hydrogel [94]. MSNs could be considered as non-swellable spherical sample. In this context, the drug-release mechanism of PAA-coated and 5-FU-loaded MCM-41 with Ca^2+^ can be better understood by using the Korsmeyer–Peppas model. 

To confirm the release mechanism, the drug-release results were fitted to the following Korsmeyer–Peppas equation [94]:$$\frac{M_{t}}{M_{\infty }}=k_{p}t^{n}$$

In this equation, *M_t_*/*M**_∞_* is a fraction of the drug release of 5-FU at time *t*, *k_p_* is the rate constant, and *n* is the release exponent. By using the least-squares procedure, the values of *n* and *k* were estimated for all formulations as shown in Table 3. The *n* values were used to characterize the release mechanism of the samples (MCM-41_5-FU, MCM-41-PAA_5-FU and Ca@MCM-41-PAA_5-FU). For *n* = 0.5, the drug diffuses according to Fickian type diffusion. If 0.5 < *n* < 1, anomalous or non-Fickian drug diffusion works. In the case of *n* = 1, case II transport of drug kinetics occurs. In this work, the *n* values were under *n* < 0.5 (between 0.09 and 0.3), assuring a Fickian diffusion trend, which approves non-swellable property of MCM-41 nanostructure. The in vitro release results imply that these nanocarriers can be applied for the delivery of anticancer drugs.

All samples showed a burst release at the beginning of drug release (within a short time of 1 h). The result may be due to the fact that drug molecules present outside and at the entrance to the mesopores are rapidly released. After that time, the drug is released slightly slowly (until the drug-release time of 24 h for 5-Fu-loaded MCM-41 (MCM-41_5-FU), until the drug release time of 48 h for PAA-functionalized and 5-FU loaded MCM-41 (MCM-41-PAA_5-FU) and PAA-functionalized and 5-FU loaded MCM-41 with Ca^2+^ (Ca@MCM-41-PAA_5-FU)), whereas the drug release reaches equilibrium due to the long release time (longer than 48 h). Therefore, although it is difficult to perfectly match the data of this study, if it is considered to be in the range of 3–24 h for MCM-41_5-FU and in the range of 3–48 h for MCM-41-PAA_5-FU and Ca@MCM-41-PAA_5-FU), the Korsmeyer–Peppas model will be helpful in understanding the sustained drug release behavior.

## 4. Conclusions

Polyacrylic acid-functionalized MCM-41 was synthesized by free radical polymerization by using the acrylic acid monomer TMSPM as a linker material, CTABr as structure-directing agents, and TEOS as a silica precursor. The successful functionalization of PAA on the surface of MCM-41 by the analysis of various characterization measurements. After that, the drug release was investigated and confirmed by using 5-FU as a model anticancer drug with incorporating calcium ions. As a result, the carboxyl group of the outer surface of nanoparticles ionically incorporated with calcium ions acted as a gate of the pores, which showed remarkable pH-responsive activity as the result of 5-FU release profiles in pH 5.4 and 7.4. Disassociation of the ionic bond of calcium ion-polymeric layer in the functionalized MCM-41 (Ca@MCM-41-PAA_5-FU) helped to proceed pH-stimulus selective release of 5-FU and calcium ions at pH 5.4. The total drug release amount was roughly two times higher at pH 5.4 than under a neutral environment (pH 7.4). The highly released 5-FU can act as a cancer treatment agent. Meanwhile, at pH 7.4, the gate of nanoparticles remained closed and the drug release rate is not high. By these results, it is predicted that this gating mechanism has the possibility to be applied to various medical treatments by its specific pH responsiveness. Also, the stimuli-responsive property of synthesized material shows potential as an effective delivery system for other applications. Meanwhile, based on the results obtained through this study, it is considered necessary to do further in-depth studies of the influence of various other metal ions in addition to Ca^2+^ ions and biomolecules on the controlled release of drug molecules in biological conditions as well as more detailed studies on the cytotoxicity and in vitro or in vivo biological activities of such metal ions including calcium ions which may induce apoptosis of cancer cells in the body.

## Data Availability

Not applicable.

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
