# Peer review of "pH-Sensitive Polyacrylic Acid-Gated Mesoporous Silica Nanocarrier Incorporated with Calcium Ions for Controlled Drug Release"

_materials, 2022, doi:10.3390/ma15175926_

Round 1
Author Response
We thank you a lot for your valuable comments and suggestions. We did our best to incorporate your valuable suggestions and comments in this revised manuscript. We believe the quality of this revised manuscript has been significantly improved thanks to your precious comments and suggestions. We hope that our revision would be satisfactorily done. Our revision is marked in red in this revised manuscript.
Comment 1. At the moment there are many language errors and inconsistencies which need to be corrected (e.g. lines 46, 56, 108 etc.). The language of the manuscript must be considerably improved.
Answer) Thanks for your kind comment. We have improved the language as follows.
We changed ‘The pH-sensitive material’s delivery system, which can do the targeted drug release at a low pH region…’ to ‘The pH-sensitive drug delivery system, which can do the selective release at a low pH region…’. (Line 46~47)
We have correction of Line 56 while the sentence was modified. (In general, it is known that the size of…) (Line 83)
We have deleted a space between ‘…hours.’ and ‘The molar…’. (Line 146)
In addition, we have corrected some language errors and made some modifications to clarify the sentences as follows.
We changed ‘5-fluorouracil (5-FU) encapsulated mesoporous silica delivery system matrix’ to ‘5-fluorouracil (5-FU) encapsulated mesoporous silica’. (Line 110)
We changed ‘polyacrylic acid (PAA) chains’ to ‘polyacrylic acid (PAA)-Ca2+ complexes’. (Line 111)
We changed ‘metal complexes’ to ‘metal ion complexes’. (Line 119)
We changed ‘calcium atom’ to ‘calcium ions’. (Line 120)
Comment 2. It is misleading to state that the synthesized nanocarriers are “targeted nanocarriers”, or have a potential as targeted nanocarriers, as stated in lines 23 and 47, since no active targeting was accomplished (e.g. by binding of receptor-specific ligands).
Answer) Thanks for your kind comments. Based on your comments, we have revised those as follows.
We changed ‘…as pH-responsive targeted nanocarriers…’ to ‘…as pH-responsive nanocarriers…’. (Line 23)
We changed ‘The pH-sensitive material’s delivery system, which can do the targeted drug release at a low pH region…’ to ‘The pH-sensitive drug delivery system, which can do release at a low pH region…’. (Line 46~47)
Comment 3. The introduction lacks of a proper description of the current state of the art, especially with regard to mesoporous carrier particles/materials. In addition, a clear distinction should be made from existing MCM-41-based pH-responsive carrier systems and the need for the carrier system developed here should be demonstrated more clearly.
Answer) Thanks for your comments. Based on your advice, we described the studies on the drug delivery based on the MCM-41 and the drug release behavior by various stimuli including pH, reported in previous works. The drug delivery studies of 5-FU loaded mesoporous silica by pH stimuli in previous works were also described and the originality of our work was more highlighted. {(‘The pore walls or the surfaces of MSNs were… was induced by pH stimuli.’ (Line 57-66), ‘In general, it is known… by acidity have rarely been reported.’ (Line 83-110)}
Comment 4. Although already an extensive list of references is provided, several statements in the manuscript need to be supported by appropriate references (e.g. line 41 “many kinds of inorganic nanoparticles” as well as statements in lines 51-52; 64-65; 75-76; 78-79; 424-426).
Answer) Thanks for your comments. We added appropriate references to the statements based on your comments as follows.
We added references ‘20’ and ‘21’ for ‘...many kinds of inorganic nanoparticles’. (Line 41-42)
We added reference ‘28’ for ‘It has great biocompatibility and high loading capacity among other systems too.’ (Line 51-52) and ‘However, drug-polymer conjugates have bad stability and loading capacities are low.’. (Line 73-74)
We added reference ‘67’ for ‘5-FU is an anticancer drug that can be used in the treatment of stomach cancer, pancreatic cancer, and so on.’. (Line 113-114)
We added reference ‘68’ for ‘PAA has low toxicity and provides more hydrophilicity to mesoporous silica materials.’. (Line 116-117)
We added reference ‘91’ for ‘The open pore state caused by this behavior of the polymer allows 5-FU to migrate to the exit of the pore and release into the solution.’. (Line 465-466)
All added references were listed in the ‘References’ section. The references were renumbered.
Comment 5. Abbreviations need to be introduced (lines 132, 134).
Answer) According to your comment, we have changed those as follows.
We changed ‘MBA’ to ‘N,N′-methylenebis(acrylamide) (MBA)’. (Line 176 and 177)
We changed ‘KPS’ to ‘potassium persulfate (KPS)’. (Line 178)
Comment 6. Section 3.1 does not include any results nor discussion and should be deleted and Scheme 1 moved to section 2.
Answer) As your comment, we delete Section 3.1. And Scheme 1 moved to Section 2.2. A related text (‘The synthesis process was briefly shown in Scheme 1.’) has been added to the sections (Section 2.2, 2.3, 2.4, and 2.5) related to Scheme 1. (Line 149, 167, 182, and 195-196) Accordingly, the other sections thereafter have been renumbered.
Comment 7. For quantification by TGA in Figure 6, it was stated that “the initial weight loss under 200 °C is caused by thermos-desorption of physisorbed water or residual solvents”. This would mean that the increase in mass loss of curve (d) would be due to increased adsorbed solvent and not to loading of 5-FU. Or is the decomposition of the 5-FU expected to be superimposed on the desorption of the solvent? This needs to be discussed in more detail.
Answer) Thanks for your kind comments. As your comments, we have discussed it in more detail with additional explanation.
In relation to the explanations, we added several sentences to the manuscript;
“(Ca@MCM-41-PAA_5-FU probably contains a higher amount of H2O than other samples because it contains calcium ions and carboxyl groups with high hydrophilicity. Meanwhile, 5-FU may exhibit weight loss due to some decompositions as it is heated up to 200 °C. These factors will lead to a significant weight loss of Ca@MCM-41-PAA_5-FU below 200 °C than other samples.) (Line 319-323)
Comment 8. The wavenumbers mentioned in lines 313 and 314 differ from those in figure 7.
Answer) Thank you for your kind comment. We changed ‘1708’ to ‘1704’ and ‘1719 cm-1’ to ‘1715 cm-1’, respectively, in the text of the manuscript. (Line 352 and 353)
Comment 9. The claimed difference in line 318 (in figure 7) appears to be not significant in the presented diagrams.
Answer) As your comment, the description of the FTIR spectrum for Ca@MCM-41-PAA_5-FU (Figure 7d) was changed as follows.
We changed ‘Typical O–C–O stretching adsorption peaks appear at the wavelength near 1578 and 1402 cm-1. When Ca2+ ions existed, the pair of bands was observed to shift about 10 cm-1 respectively. Therefore, changes in the position of the carboxylate group’s adsorption peaks display that the polymer-metal ion complex on the surface of MCM-41 was formed by the interaction between carboxylate groups and the metal ions [63].’ to ‘The peaks were slightly shifted due to the interaction between calcium ions and carboxyl groups compared to the MCM-41-PAA without calcium ions [82].’. (Line 358-359)
Comment 10. The isotherms of (b) and (c) in figure 8 cannot be classified as type III isotherms but rather as type I isotherms, which are characteristic for microporous materials as the obtained MCM‐41‐ PAA and (c) Ca@MCM‐41‐PAA_5‐FU (pore size < 2 nm; see table 1).
Answer) Thanks for your kind comment. As your comment, we changed ‘Type III’ to ‘Type I’. (Line 380)
Comment 11. The synthesized particle systems have a hydrodynamic size of 335 nm (figure 9), which is too large for endocytosis-mediated intracellular internalization and thus does not meet the criterion of a maximum size of 100 nm (as also described in the introduction). This result needs to be discussed in more detail as to how an application could still be realized.
Answer) Thanks for your comment. In general, it is known that the size of nanoparticles of about 100 nm is most suitable for delivery to cells by endocytosis. However, it has been reported that mesoporous silica with a large particle size can also be sufficiently applied as a carrier for drug delivery in cells. Moorthy et al. [37] reported on the research of the simultaneous cell imaging and the anticancer drug delivery in MCF-7 cells with the safranin–diurea bridged and 5-fluorouracil (5-FU) loaded mesoporous organosilica (5-FU loaded SDU-HMS). The 5-FU loaded SDU-HMS showed excellent cytotoxicity to the MCF-7 cells with cell viability of approximately 18 %. The mesoporous silica was a particle size in the range of 200-800 nm. Moorthy et al. [63] reported also the synthesis of the carboxyl- (DU-MSH-COOH) and amine-functionalized mesoporous silica (DU-MSH-NH2) with a particle size in the range of 500 nm-1.5 μm. The internalized 5-FU loaded DU-MSH-COOH and DU-MSH-NH2 into MCF-7 cells showed excellent cytotoxicity of 76 % and 82 %, respectively. Tao et al. [64] reported on the endocytosis and the time-dependent enhanced cytotoxicity of anticancer platinum drug-loaded MCM-41 (particle size in the range of 500-900 nm) or SBA-15 (particle size of the micrometer). The mesoporous silica particles with large sizes may be suitable for the cells with comparatively large sizes such as MCF-7 and HeLa cells [37].
The above-mentioned contents have been added in the introduction section. (In general, it is known… such as MCF-7 and HeLa cells [37].) (Line 83-98) And references related to its contents were added as references 37, 63 and 64, and were listed in the ‘References’ section.
As new references were added, other references were renumbered.
And we added a sentence for description in detail the hydrodynamic size of the sample. (‘The hydrodynamic diameter of the material was 349.7 (pH 7.4) and 394.0 nm (pH 5.4).’) (Line 111-112)
Comment 12. The discussion of the results in section 3.2.9 lacks scientific depth. In addition, the question arises as to why only a maximum release of approx. 50% is achieved in Figure 10 B (a) and C (b). This also requires an explanation.
Answer) At a neutral pH environment (pH 7.4) (Figure 10Bb), carboxylic acid groups can alter their ionization/deprotonation behavior depending on the pH environment. Most of the carbonyl groups of polyacrylic acid (pKa = 4.8) are negatively charged at pH 7.4 which makes electrostatic repulsion between polymer chains. Also, its high solubility results in PAA chains’ swelling behavior. The open pore state caused by this behavior of the polymer allows 5-FU to migrate to the exit of the pore and release into the solution. Note that there is no significant interaction between 5-FU (pKa = 8.2) and carboxyl groups [63]. In contrast, in a pH 5.4 environment (Figure 10Ba), the contraction of PAA chains is contributed by hydrogen bonding between carboxyl groups by protonation. The result could block the pore entrances so the release of 5-FU would be hindered [90]. Meanwhile, it showed a rapid release behavior of 5-FU with a short time of less than 1 h. This result may be attributed to the release of drug molecules present outside and at the entrance to the mesopore. A similar release behavior was also shown in Figure 10C.
Compared to the result of Figure 10B, in Figure 10C, the tendency of drug release amount was changed. The total release amount in pH 5.4 and pH 7.4 was 93.7% and 54.1%, respectively. In a neutral environment (pH 7.4) (Figure 10Cb), the carboxylate groups with negative charges could form COO−−Ca2+ electrostatic interaction. This ionic bond makes a polymer-metal ion complex layer around MCM-41 nanoparticles, so the release of 5-FU would be obstructed. At pH 7.4, however, some of the 5-FU molecules can be deprotonated [92], then the interaction between 5-FU and calcium ion could be formed that can break the COO−−Ca2+ interaction. This disassociation of polymer-metal ion bond could arise leakage of 5-FU drugs contained in mesopores. At pH 5.4, the interacted polymer-calcium ion layer was disbanded and the drug release amount increased. The protonation of carboxyl groups at comparably low pH leads to withdraw electrostatic interaction between COO− and Ca2+. This disruption of the polymer-metal ion layer led to the obvious release of 5-FU drugs into the solution. Also, Cl− ions in PBS buffer support the formation of calcium chloride and replace carboxylate groups, thereby disrupting the ionic interaction between COO− and Ca2+ [93]. By these results, the Ca2+ interacting polymeric layer could act as a pH-selective drug release barrier and inhibit burst release in a normal physiological environment.
Most of the contents described above have already been described in the text part explaining Figure 10. But, as your comments, the scientific explanation of the results in Figure 10 was supplemented, while the sentences were modified to clarify the contents.
The supplements and corrections are as follows.
We changed ‘At neutral pH condition’ to ‘At neutral pH condition (pH 7.4)’. (Line 447)
And we added the following sentences: ‘MCM-41_5-FU showed a release amount of about 67% within a short time of 1 h (Figure 10A). The result can be due to the fact that drug molecules present outside and at the entrance to the mesopores are rapidly released. After that time, the drug is released slightly slowly until the drug release time of 24 h, and this result may be attributed to the route of release from the inside of the mesopores to the outside.’ (Line 450-454)
We changed ‘At a neutral pH environment’ to ‘At a neutral pH environment (pH 7.4) (Figure 10Bb)’. (Line 461)
We changed ‘The opened pore state caused by this behavior of polymers made 5-FU yielding to moving to the exit of pores and the release in solutions.’ to ‘The open pore state caused by this behavior of the polymer allows 5-FU to migrate to the exit of the pore and release into the solution.’ (Line 465-466)
We changed ‘in a pH 5.4 environment, contraction of PAA chains by protonation could block the pore exit so the release of 5-FU would be hindered [74].’ to ‘in a pH 5.4 environment (Figure 10Ba), the contraction of PAA chains is contributed by hydrogen bonding between carboxyl groups by protonation. The result could block the pore entrances so the release of 5-FU would be hindered [90].’ (Line 468-470)
We added the following sentences: ‘Meanwhile, it showed a rapid release behavior of 5-FU with a short time of less than 1 h. This result may be attributed to the release of drug molecules present outside and at the entrance to the mesopore. A similar release behavior was also shown in Figure 10C.’ (Line 470-473)
We changed ‘In a neutral environment (pH 7.4),’ to ‘In a neutral environment (pH 7.4) (Figure 10Cb),’. (Line 476)
We changed ‘Cl− ion in PBS buffer could replace the carboxylate groups to destroy the ionic interaction between COO− and Ca2+ [76].’ to ‘Cl− ions in PBS buffer support the formation of calcium chloride and replace carboxylate groups, thereby disrupting the ionic interaction between COO− and Ca2+ [93].’ (Line 486-488)
Comment 13. A deeper explanation on the Korsmeyer-Peppas equation would be beneficial for an improved understanding of the model.
Answer) Korsmeyer-Peppas model is used to describe and analyze the release of a drug from a polymeric nanoparticles dosage form such as a hydrogel [95]. MSNs could be considered as non-swellable spherical sample. In this context, the drug release mechanism of PAA-coated and 5-FU-loaded MCM-41 with Ca2+ can be well understood using the Korsmeyer-Peppas model.
To confirm the release mechanism, the drug release results were fitted to the following Korsmeyer-Peppas equation [90].
In this equation, Mt/M∞ is a fraction of the drug release of 5-FU at time t, kp is the rate constant, and n is the release exponent. The n values were used to characterize the release mechanism of the samples (MCM-41_5-FU, MCM-41-PAA_5-FU and Ca@MCM-41-PAA_5-FU). For n = 0.5, the drug diffuses according to Fickian type diffusion. If 0.5 < n < 1, anomalous or non-Fickian drug diffusion works. In the case of n = 1, case II transport of drug kinetics occurs. In this work, the n values were under n < 0.5 (between 0.09 and 0.3), assuring a Fickian diffusion trend, which approves non-swellable property of MCM-41 nanostructure.
Most of the content mentioned above has already been described in the manuscript. Additionally, a few sentences have been added. (‘Korsmeyer-Peppas model is used to describe and analyze the release of a drug from a polymeric nanoparticles dosage form such as a hydrogel [95]. MSNs could be considered as non-swellable spherical sample. In this context, the drug release mechanism of PAA-coated and 5-FU-loaded MCM-41 with Ca2+ can be well understood using the Korsmeyer-Peppas model.’) (Line 502-506)
And a reference related to its contents was added as Ref. ‘95’ and was listed in the ‘References’ section. The references were renumbered.
The corrected parts are marked in red. Many thanks again for your valuable comments and suggestions.
Reviewer 2 Report
Figure 5A. Are release curves significantly different?
Silanol, PAA and 5-FU are weak acids. I would expect higher release at higher pH.
There is clearly a two phase release behavior. A burst release and a sustained release. The Korsmeyer‐Peppas equation can not be applied for all the sampling data. That’s the reason for n<0.5, the data doesn’t fit the model. The sustained release can be diffusion controlled but is not demonstrated with the analysis performed.
How the authors control the formation of a hydrogel?
The novelty of the proposed system is the use of Ca ions to control the release. In biological systems Ca ions can be exchanged by more abundant ions, and sequestered by carrying molecules such as albumin. In such can it may not be a proper control.
Author Response
We thank you a lot for your valuable comments and suggestions. We did our best to incorporate your valuable suggestions and comments in this revised manuscript. We believe the quality of this revised manuscript has been significantly improved thanks to your precious comments and suggestions. We hope that our revision would be satisfactorily done. Our revision is marked in red in this revised manuscript.
Q1. Figure 5A. Are release curves significantly different?
Answer) Even taking into account experimental errors in the data, we believe that the difference in the release results performed under each condition are significant within experimental errors. In order to display the significance in the experimental data, we added error bars in Figure10.
Q2. Silanol, PAA and 5-FU are weak acids. I would expect higher release at higher pH.
Answer) In this work, a higher release amount is observed under acidic condition (pH 5.4) than neutral condition (pH 7.4) for the MCM-41_5-FU due to the partial protonation of 5-FU (pKa=8.2) and silanol groups on the surface of pure MCM-41 that cause electrostatic repulsion of the drug molecules under lower pH condition (pH 5.4) [63]. Meanwhile, at neutral pH condition (pH 7.4), both hydrophilic 5-FU and silanol groups made a hydrophilic-hydrophilic interactions and hydrogen bonding could be formed so that the release rate of 5-FU drugs could be slower [63].
In the case of 5-FU loaded and carboxyl group functionalized MCM-41 (MCM-41-PAA_5-FU), however, at a neutral pH environment (pH 7.4), carboxylic acid groups can alter their ionization/deprotonation behavior depending on the pH environment. Most of the carbonyl groups of polyacrylic acid (pKa = 4.8) are negatively charged at pH 7.4 which makes electrostatic repulsion between polymer chains. Also, its high solubility leads to PAA chains’ swelling. The open pore state caused by this behavior of the polymer allows 5-FU to migrate to the exit of the pore and release into the solution [91]. Note that there is no significant interaction between 5-FU (pKa = 8.2) and carboxyl groups [63]. In contrast, in a pH 5.4 environment, the contraction of polyacrylic acid (PAA) chains is contributed by hydrogen bonding between carboxyl groups by protonation. The result could block the pore entrances so that the release of 5-FU would be hindered [91].
With referring two new references (Ref. 63 and 91), we added such further discussion in this revised manuscript with the pKa value of 5-FU, since the content described above was already well described in those references (Ref. 63 and 91) as well as in our original manuscript. (‘Meanwhile, it shows a higher release amount …5-FU drugs could be slower’ (Line 445-450) and ‘In the case of 5-FU loaded MCM-41-PAA (MCM-41-PAA_5-FU)… of 5-FU would be hindered [91]’ (Line 460-470))
Q3. There is clearly a two phase release behavior. A burst release and a sustained release. The Korsmeyer‐Peppas equation can not be applied for all the sampling data. That’s the reason for n<0.5, the data doesn’t fit the model. The sustained release can be diffusion controlled but is not demonstrated with the analysis performed.
Answer) Thanks for your very good comments. All samples showed a burst release at the beginning of drug release (within a short time of 1 h). The result may be due to the fact that drug molecules present outside and at the entrance to the mesopores are rapidly released. After that time, the drug is released slightly slowly {until the drug release time of 24 h for 5-FU loaded MCM-41 (MCM-41_5-FU), until the drug release time of 48 h for PAA-functionalized and 5-FU loaded MCM-41 (MCM-41-PAA_5-FU) and PAA-functionalized and 5-FU loaded MCM-41 with Ca2+ (Ca@ MCM-41-PAA_5-FU)}, while the drug release reaches equilibrium due to the long release time (longer than 48 h). Therefore, although it is difficult to perfectly match the data of this study, if it is considered in the range of 3-24 h for MCM-41_5-FU and in the range of 3-48 h for MCM-41-PAA_5-FU and Ca@MCM-41-PAA_5-FU), the Korsmeyer-Peppas model will be helpful in understanding the sustained drug release behavior.
The statements mentioned above described in the manuscript. (All samples showed a burst release… in understanding the sustained drug release behavior.) (Line 524-534)
Q4. How the authors control the formation of a hydrogel?
Answer) As described in detail in the 'Experimental Section' of the manuscript, first, alkoxysilane with double bond groups is modified in the synthesized mesostructured silica. In this way, organic functional groups with double bond groups are modified at the entrance of the nanopores. And the template (CTABr) contained in the nanopore is removed by the solvent-extraction method. Then, polyacrylic acid-functionalized MCM-41 is synthesized by free-radical polymerization of acrylic acid with an organic group having a double bond in mesoporous silica. Finally, polyacrylic acid (PAA) is modified at the entrance of the nanopores. The PAA polymer chain can be shrunk (blocking the nanopores and thus inhibiting drug release) or swell (opens nanopores and releases drug molecules). The result occurs through the protonation or deprotonation of the carboxyl groups in the controlled pH. When calcium ion is added to polyacrylic acid-functionalized MCM-41, the calcium ion interacts with the carboxyl groups in the PAA chain to form a network. The interaction between calcium ions with carboxyl groups in the PAA chain can be controlled by pH. When the interaction between calcium ions and carboxyl groups is strong, the PAA polymer-calcium ion network is well formed and the release of drug molecules in the nanopore is disturbed. On the other hand, when the interaction between the calcium ion and the carboxyl groups is weak, the PAA chain swells in the solution and the drug molecule is released.
The above-described contents are explained in detail in the sections 2.3. - 2.5. in the 'Experimental Section'. Also, we moved Scheme 1 in the Experimental section with appropriate sentences from the section 3.1. in our original manuscript, which might be more clear to understand the formation of hydrogel. (Line 149, 167, 182, 195-196)
Q4. The novelty of the proposed system is the use of Ca ions to control the release. In biological systems Ca ions can be exchanged by more abundant ions, and sequestered by carrying molecules such as albumin. In such can it may not be a proper control.
Answer) Thanks for your very kind comments. We would like to do more in-depth research under various conditions on the things you mentioned in the future. Accordingly, we briefly mentioned this statement in the revised manuscript; We added a sentence, ‘Meanwhile, based on the results obtained through this study, it is considered necessary to do further in-depth study the influence of various other metal ions in addition to Ca2+ ions and biomolecules on the controlled release of drug molecules in biological conditions.’ in ‘Conclusions’ section. (Line 553-556)
Many thanks again for your valuable comments and suggestions.
Reviewer 3 Report
The manuscript by Kong et al reports the pH‐Sensitive Polyacrylic Acid‐Gated Mesoporous Silica Nanocarrier Incorporated with Calcium Ions for Controlled Drug Release. The manuscript presents details of the methods used and their properties. The detail and clarity in the methods are good.
General comments:
1. Manuscript needs moderate English spelling editing and grammar checks.
- The introduction section needs improvement. Please mention the methods used in the manuscript.
- Authors should review the text and correct any editing (ml) errors.
- The result and discussion part can be strengthened with more references. In particular, the results part of the drug releasing test can be strengthened with further discussion and reference.

Author Response
We thank you a lot for your valuable comments and suggestions. We did our best to incorporate your valuable suggestions and comments in this revised manuscript. We believe the quality of this revised manuscript has been significantly improved thanks to your precious comments and suggestions. We hope that our revision would be satisfactorily done. Our revision is marked in red in this revised manuscript.
The manuscript by Kong et al reports the pH‐Sensitive Polyacrylic Acid‐Gated Mesoporous Silica Nanocarrier Incorporated with Calcium Ions for Controlled Drug Release. The manuscript presents details of the methods used and their properties. The detail and clarity in the methods are good.
Answer: Thank you very much for your encouragement.
General comments:
1.Manuscript needs moderate English spelling editing and grammar checks.
Answer: Thanks for your kind comments. According to your kind advice, we carefully checked grammatical and typoerrors and corrected, if any, throughout the text.,
2.The introduction section needs improvement. Please mention the methods used in the manuscript.
Answer; We added several sentences in the introduction to highlight the novelty of our manuscript in more details as follows;
Based on your advice, we described the studies on the drug delivery based on the MCM-41 and the drug release behavior by various stimuli including pH, reported in previous works. The drug delivery studies of 5-FU loaded mesoporous silica by pH stimuli in previous works were also described and the originality of our work was more highlighted. {(‘The pore walls or the surfaces of MSNs were… was induced by pH stimuli.’ (Line 57-66), ‘In general, it is known… by acidity have rarely been reported.’ (Line 83-110)}
In addition, we added appropriate references to support statements in the manuscript as follows.
We added references ‘20’ and ‘21’ for ‘...many kinds of inorganic nanoparticles’. (Line 41-42)
We added reference ‘28’ for ‘It has great biocompatibility and high loading capacity among other systems too.’ (Line 51-52) and ‘However, drug-polymer conjugates have bad stability and loading capacities are low.’. (Line 73-74)
We added reference ‘67’ for ‘5-FU is an anticancer drug that can be used in the treatment of stomach cancer, pancreatic cancer, and so on.’. (Line 113-114)
We added reference ‘68’ for ‘PAA has low toxicity and provides more hydrophilicity to mesoporous silica materials.’. (Line 116-117)
We added reference ‘91’ for ‘The open pore state caused by this behavior of the polymer allows 5-FU to migrate to the exit of the pore and release into the solution.’. (Line 465-466)
All added references were listed in the ‘References’ section. The references were renumbered.
The methods are mentioned in the section 2. Experimental section from Line 130 -235 in detail.
3.Authors should review the text and correct any editing (ml) errors.
Answer; We corrected the editing (ml) errors as “mL” throughout the text. We are sorry for the mistakes.
4.The result and discussion part can be strengthened with more references. In particular, the results part of the drug releasing test can be strengthened with further discussion and reference.
Answer; The results and discussion were more strengthened with more discussion as follows; To support our discussion, we added new references [63] and [91] as well as [37] and [63] in the Introduction.
For quantification by TGA in Figure 6; we have discussed it in more detail with additional explanation;
“(Ca@MCM-41-PAA_5-FU probably contains a higher amount of H2O than other samples because it contains calcium ions and carboxyl groups with high hydrophilicity. Meanwhile, 5-FU may exhibit weight loss due to some decompositions as it is heated up to 200 °C. These factors will lead to a significant weight loss of Ca@MCM-41-PAA_5-FU below 200 °C than other samples.) (Line 319-323)
Further discussion of the results in section 3.1.9 (Drug Delivery Behavior);
“MCM-41_5-FU showed a release amount of about 67% within a short time of 1 h (Figure 10A). The result can be due to the fact that drug molecules present outside and at the entrance to the mesopores are rapidly released. After that time, the drug is released slightly slowly until the drug release time of 24 h, and this result may be attributed to the route of release from the inside of the mesopores to the outside.’ (Line 450-454)
‘Meanwhile, it showed a rapid release behavior of 5-FU with a short time of less than 1 h. This result may be attributed to the release of drug molecules present outside and at the entrance to the mesopore. A similar release behavior was also shown in Figure 10C.’ (Line 470-473)
More deeper explanation on the Korsmeyer-Peppas equation ;
‘Korsmeyer-Peppas model is used to describe and analyze the release of a drug from a polymeric nanoparticles dosage form such as a hydrogel [95]. MSNs could be considered as non-swellable spherical sample. In this context, the drug release mechanism of PAA-coated and 5-FU-loaded MCM-41 with Ca2+ can be well understood using the Korsmeyer-Peppas model.’) (Line 502-506)
The corrected part is marked in red text. Many thanks for your valuable comments again.
Round 2
Reviewer 1 Report
- The entire paper should be rechecked for language errors or phrasing (such as in line 44, 408/409, 451, and more).
- The introduced explanation regarding the particle size should not be added in the introduction section but rather as a discussion to the obtained results in section 3.1.8.
- The phrasing in line 547/548 that the "calcium ions can induce apoptosis of cancer cells in the body" should be revised since no experiments in the presented work have been conducted which dealt with the cytotoxicity of the calcium ions. Such a phrase could be used as a potential outlook.
Author Response
We thank you a lot for your valuable comments and suggestions. We did our best to incorporate your valuable suggestions and comments in this revised manuscript. We believe the quality of this revised manuscript has been significantly improved thanks to your precious comments and suggestions. We hope that our revision would be satisfactorily done. Our revision is marked in red in this re-revised manuscript.
- The entire paper should be rechecked for language errors or phrasing (such as in line 44, 408/409, 451, and more).
Answer: We are sorry for such careless mistakes in English. We rewrote some phrases after checking all the sentences in the manuscript as follows;
In Line 42-43; . The original sentence was; “ In response to external stimuli (temperature, light irradiation, pH, redox, etc.), the release amount of the drug can be controlled more precisely than the drug without any device.
Revised one; “The release amounts of drugs can be controlled in response to external stimuli (temperature, light irradiation, pH, redox, etc.).”
In Line 408-409; The original sentence was; “ It might be less affected by the release of drugs which is less amount compared to the release of MCM-41_5-FU.”
Revised one; “ It might be less affected by the release of drugs, since the release amount from MCM-41-PAA_5-FU is less than that from MCM-41_5-FU.”
In Line 451; The original one was; “The result can be due to the behavior in which drug molecules present outside and at the entrance to the mesopores are rapidly released.”
Revised one; “The result can be due to the fact that drug molecules present outside and at the entrance to the mesopores are rapidly released.”
In addition,
In Line 51; The original one was; “It has great biocompatibility and high loading capacity among other systems too [28].”
Revised one; “It also has great biocompatibility and high loading capacity among other systems [28].”
In Line 107; “5Fu” was corrected to “5-FU”.
In Line 316; “10 °Cmin-1 “was corrected to “10 °C min-1.“
In Line 354; “more obvious” was corrected to “higher”.
In Line 382-383; The original sentence was; “Meanwhile, for the pore size of Ca@MCM-41-PAA_5-FU, it was too low value to measure (< 2.0 nm).”
Revised one ; “ Meanwhile, the pore size of Ca@MCM-41-PAA_5-FU was too small to measure (< 2.0 nm).”
In Line 397 ; “neutral” was corrected to “more neutral.”
In Line 399; “obvious” was corrected to ‘distinct.”
- The introduced explanation regarding the particle size should not be added in the introduction section but rather as a discussion to the obtained results in section 3.1.8.
Answer: Based on your kind suggestion, we removed the explanation regarding the particle size from the Introduction. The particle size is already mentioned in the section 3.1.8. (Lines 428-429).
- The phrasing in line 547/548 that the "calcium ions can induce apoptosis of cancer cells in the body" should be revised since no experiments in the presented work have been conducted which dealt with the cytotoxicity of the calcium ions. Such a phrase could be used as a potential outlook.
Answer: According to your kind suggestion, we removed the phrase, “calcium ions can induce apoptosis of cancer cells in the body”, in Line 547, but instead we added it as a potential outlook which needs further studies in future in Lines 554-557 such as, “… in biological conditions as well as more detailed studies on the cytotoxicity and in vitro or in vivo biological activities of such metal ions including calcium ions which may induce apoptosis of cancer cells in the body.”
In addition, the same phrasing in Line 497 in the original manuscript was also removed.
Many thanks again for your valuable and kind suggestions and comments.